# Ethical and methodological reflections: Digital storytelling of self-care with students during the COVID-19 pandemic at a South African University

Dumile Gumede[1]*, Maureen Nokuthula Sibiya[2]

**1** Centre for General Education, Durban University of Technology, Berea, Durban, South Africa, **2** Division of Research, Innovation and Engagement, Mangosuthu University of Technology, Umlazi, Durban, South Africa

* dumileg@dut.ac.za

**Data Availability Statement:** The datasets generated and/or analysed during the current study are not publicly available due to confidentiality

## Abstract

The enforcement of the coronavirus disease 2019 (COVID-19) pandemic restrictions disrupted the traditional face-to-face qualitative data collection in public health. The pandemic forced qualitative researchers to transition to remote methods of data collection such as digital storytelling. Currently, there is a limited understanding of ethical and methodological challenges in digital storytelling. We, therefore, reflect on the challenges and solutions for implementing a digital storytelling project on self-care at a South African university during the COVID-19 pandemic. Guided by Salmon's Qualitative e-Research Framework, reflective journals were used in a digital storytelling project between March and June 2022. We documented the challenges of online recruitment, obtaining informed consent virtually, and collecting data using digital storytelling as well as the efforts of overcoming the challenges. Our reflections identified major challenges, namely online recruitment and informed consent compromised by asynchronous communication; participants' limited research knowledge; participants' privacy and confidentiality concerns; poor internet connectivity; quality of digital stories; devices with a shortage of storage space; participants' limited technological skills; and time commitment required to create digital stories. Strategies adopted to address these challenges included an ongoing informed consent process; flexible timelines for the creation of digital stories; one-on-one guidance on creating digital stories; and multiple online platforms to share digital stories. Our critical reflection offers practical guidance for the ethical conduct of digital storytelling in public health research and makes a significant contribution to methodological considerations for use in future pandemics. These ethical and methodological challenges should be recognized as features of the context of the research setting including restrictions imposed by the COVID-19 pandemic than disadvantages of digital storytelling.

agreements but data requests can be made to the research ethics committee (irec@dut.ac.za).

**Funding:** This study was supported by the National Research Foundation of South Africa (Grant No: 138175). The funding body had no role in the design of the study and collection, analysis, and interpretation of data and in writing the manuscript.

**Competing interests:** The authors have declared that no competing interests exist.

## Introduction

COVID-19 was declared a pandemic by the World Health Organization (WHO) on 11 March 2020 after assessments of the rapid spread and severity of the deadly virus across the globe [1]. It is caused by severe acute respiratory syndrome coronavirus 2 (SARS-CoV-2) which emerged in Wuhan, China [2] and has resulted in millions of infections and deaths worldwide [1]. The first case of COVID-19 disease in South Africa was reported on 05 March 2020 [3]. To mitigate the spread of the pandemic, nations across the globe invoked different public health measures such as national lockdowns, physical distancing, and hygiene practices [4]. Like the rest of the world, the South African government enforced COVID-19 restrictions including temporarily closing all educational and research institutions to prevent the spread of the virus.

Globally, the COVID-19 pandemic disrupted the traditional face-to-face qualitative research designs through restrictive public health mandates and physical distancing [5]. Scholars [6, 7] made calls for qualitative researchers to continue contributing to informing evidence-based public health responses through qualitative studies in the era of the pandemic. According to Salmons [8], qualitative inquiry focuses on understanding the meaning people give to their lived experiences. Recent work by Paulus and Lester [9, p.335] has established that "As we live through COVID-19 and the demands for physical distancing . . . researchers must re-envision their research practices with digital tools in digital spaces." The pandemic forced qualitative researchers to transition from face-to-face data collection to remote methods of data collection such as phone or internet-based [5, 10]. This shift to remote qualitative research designs impacted how data could be collected and what instruments could be used to generate data in order to capture participants' lived experiences.

Several scholars [5, 11, 12] began to attend to how qualitative researchers may leverage technology to collect data remotely, providing recommendations, for example, on the advantages and disadvantages of remote qualitative research tools for research purposes. Remote qualitative research designs have quickly gained popularity to overcome the restrictions of face-to-face data collection in the era of the COVID-19 pandemic [10]. For example, a study in Kenya reported that using online interviews was effective in the prevention of the spread of the COVID-19 virus [13]. It has been established that remote qualitative methods offer flexibility in the location of data collection [14], meaning researchers and participants can be anywhere instead of being in the same venue as compared to face-to-face methods. Additionally, employing remote methods can aid in the faster recruitment of a wide target population [12, 15] and the participation of participants across a geographically spread population [14], overcoming time in the data collection process [16]. Conversely, online recruitment has the potential of targeting only active online users in qualitative health studies thus, raising selection bias concerns [17]. Of particular importance, online data collection may hinder the participation of those who do not have devices, lack access to internet connectivity [12], or those with limited digital literacy skills [15]. Additionally, people with disabilities e.g. blind people or socio-economically poor people who do not have smartphones or the ability to read emails are likely to be excluded from participation in online studies. Moreover, observation of participants' body language can be restricted when cameras are switched off in online environments [17].

A substantial body of literature exists on remote qualitative health research [10, 17, 18]. Within the literature domain of remote qualitative health research, a plethora of articles focus on ethical and methodological issues of collecting data using online focus group discussions [16] and online interviews [13, 15]. For example, in a US study using online focus groups with low socio-economic status African American adults during COVID-19 reported challenges related to participant privacy and online connectivity due to digital divides and technology use [16]. Yet, relatively few scholarly articles highlight the ethical and methodological challenges of

using digital storytelling in qualitative health research, particularly in the context of the COVID-19 pandemic.

Digital storytelling is an arts-based methodology that facilitates the creation of 3-to-5-minute short films called digital stories using voiceovers, images, text, video, and music for individuals to tell the story of their experiences [19], as developed by Lambert [20]. Digital stories are used to empower participants to engage in personal reflections about their experiences of a given phenomenon [21]. As a qualitative research tool, digital storytelling has already been used in Africa in the fields of public health [22–25]. For instance, digital storytelling was applied to explore community experiences of antiretroviral therapy (ART) adherence in South Africa [23]. A systematic review highlighted five goals of digital storytelling in public health interventions and research: (a) digital stories as 'counter-narratives'; (b) knowledge translation; (c) preservation of cultural heritage; (d) community development; and (e) participatory research with marginalized groups [26]. While digital storytelling shows promise for public health research, there are also challenges that must be navigated [21]. Specific challenges include the quality of digital stories produced [27]; the potential of researchers shaping participants' digital stories and lack of participants' confidentiality [21]; and concerns of stigma if digital stories are shared with the public [24]. Despite the reported challenges, digital storytelling could be modified to address the ethical and logistical challenges of working with vulnerable groups [25]. Given the growing interest in digital storytelling, it warrants critical examination. However, there remains a paucity of evidence on ethical and methodological challenges in applying this method.

To address this gap, this article presents our critical reflections on conducting an entirely remote qualitative study using the digital storytelling method to explore the self-care practices of first-year university students during the COVID-19 pandemic. We critically reflect on the ethical and methodological challenges arising in this study when using the digital storytelling method with this population. We also share some of the solutions that we implemented to address the challenges of using the digital storytelling method. The ethical and methodological dimensions, considerations, and challenges that are associated with digital storytelling data collection in research involving students remain unclear.

Engaging first-year university students in public health research is crucial for health promotion interventions [28]. Often first-year university students are in the transition from adolescence to young adulthood [29], which is a high-risk period for mental illness [30, 31], alcohol misuse [32], illicit drug use [33], violence [34], unplanned pregnancies and sexually transmitted infections including HIV [35], and unhealthy eating behaviours [36]. Moreover, the emergence of the COVID-19 pandemic resulted in universities having to shift from contact to online remote teaching and learning [37, 38]. Within the context of South African universities, one group of students that were particularly impacted by remote teaching and learning imposed by the COVID-19 pandemic was first-year students [38]. This is on account that the first year of university study is regarded as the foundation of the higher education journey which determines the educational outcomes of students [39]. According to previous studies [39, 40]. transitioning to higher education institutions can be traumatic and demanding for first-year students. Despite the adjustment challenges in a new tertiary environment [41, 42], the remote teaching and learning and the COVID-19 pandemic were unanticipated and disrupted social lives for South Africa's students [38, 39].

Therefore, understanding the ethical and methodological issues of digital storytelling among the study population of first-year university students can inform the design of public health studies using the digital storytelling method. Such research can inform how more diverse university students can be engaged in digital storytelling, given that youths have been found to be active users of online platforms [43, 44].

## Conceptual framework

Grounded in Salmon's Qualitative e-Research Framework [8], which positions online qualitative research as a critical reflection by researchers from research design throughout all the steps of the research process. The Qualitative e-Research Framework [8] shown in Fig 1 provides a holistic, circular system of interrelated steps of online qualitative research, from aligning purpose and design to analysing the data and reporting. It is beyond the scope of this article to reflect on the steps of Salmon's framework. We focus on handling sampling and recruiting, addressing ethical issues, and collecting the data, in guiding our reflections on using the digital storytelling method to explore students' self-care during the COVID-19 pandemic. Accordingly, the reflections here are anticipated to hold value for public health scholars seeking to employ the digital storytelling method in remote qualitative research.

Therefore, the intent of this article is to discuss the implementation of a digital storytelling study and present some of the ethical and methodological challenges of this study design and potential solutions from our perspectives as researchers. We share our collective insights into conducting a digital storytelling study with first-year university students. To our knowledge,

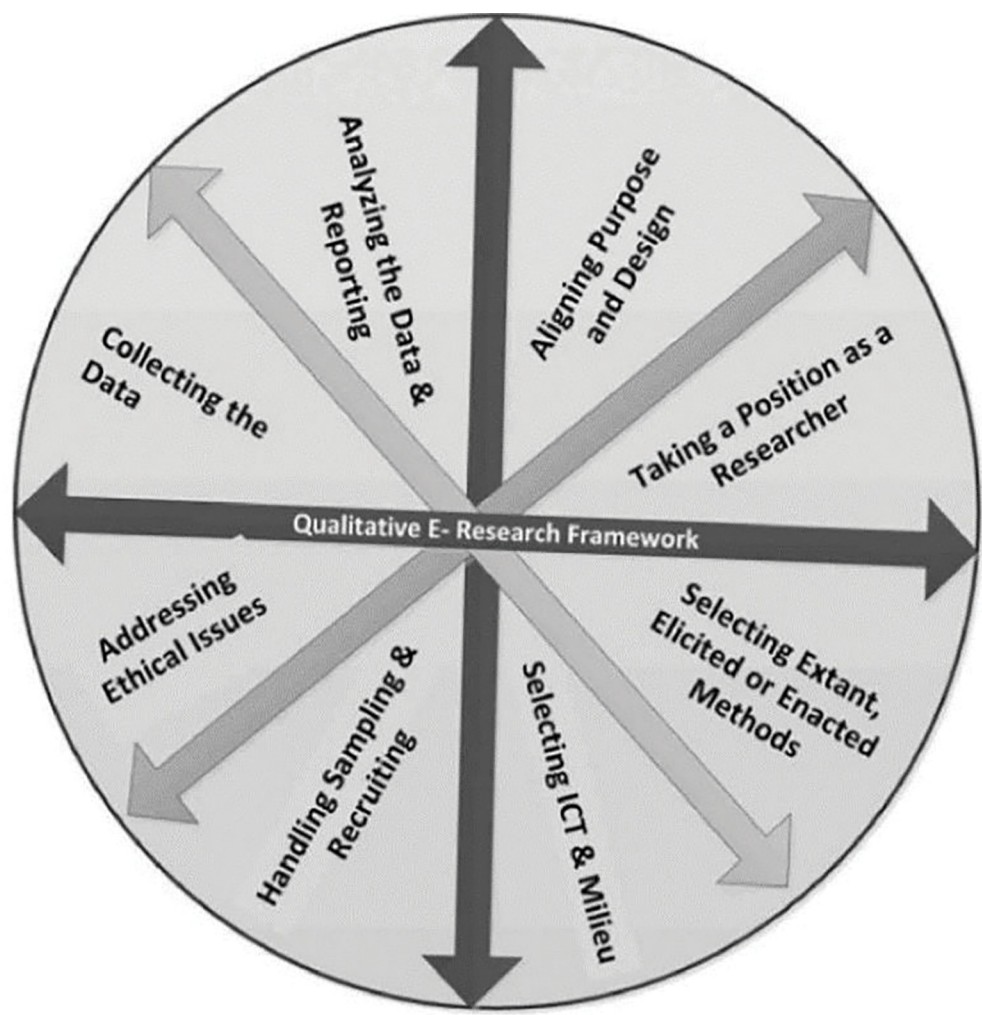

**Fig 1. Salmon's Qualitative e-Research Framework [8].**

this project is the first digital storytelling study on self-care with first-year university students to collect data completely remotely in South Africa. While our reflections are specific to first-year university students, our insights may be useful to other younger populations.

## Methods

We situate our discussion of remote qualitative research using the digital storytelling method in our work investigating first-year students' self-care at a university in South Africa. As part of Phase 1 of a longitudinal project, we examined students' self-care practices and the contextual factors shaping their self-care practices in the era of the COVID-19 pandemic. Our work sought to understand first-year students' self-care practices across six domains that focus on the whole person [45]: physical, professional, relational, emotional, psychological, and spiritual. Thus, it was critical to use digital stories as a qualitative research tool to show how student-authored accounts have the potential to provide valuable insights into the personal contexts of self-care during the time of the COVID-19 pandemic.

We decided to apply digital storytelling as a methodology in this project since it can be particularly relevant when working with individuals remotely. Recruitment for this study took place online to reach as diverse participants as possible so that more voices, perspectives, and experiences could be shared on how first-year students practiced self-care and the contextual issues that shaped their self-care practices. We advertised an open invitation to participate in the study called '*Digital Stories of Self-care*' via university notices and students' email accounts. With the invitation, an MS Teams link to access the study information letter and informed consent form was included. MS Teams was selected because it is the telecommunications interface that is sponsored and supported by the university's information technology department. The invitation was sent out twice in March 2022. A purposive sample technique was used to select a sample of between 20 to 30 students to participate in the study until saturation was reached as evidenced by similar responses being made by participants. Criteria for selecting the participants were as follows: first-year students who were registered for the first time in higher education in the academic year 2022. Returning students were excluded from participation in the study.

A total of 26 students (13 females and 13 males) aged between 17 and 27 years participated in the study between March and June 2022. Over and above those who participated in the study, only three participants withdrew from participation by not submitting their digital stories. The reasons for withdrawal were not captured. As participation in the digital storytelling project was completely online, participants were required to have a personal device such as a smartphone, tablet, or laptop that they could use to create their digital stories.

Unlike other scholars [23, 24], we opted to hold one-on-one sessions via MS Teams to guide participants on basic steps for creating digital stories in order to ensure the privacy and confidentiality of participants instead of running digital storytelling workshops with the participants as a group. This strategy was aimed at preventing participants from presenting self-care practices that could have been perceived as socially acceptable in a group workshop. As noted by Mnisi [22], our intention was also to allow the participants to control what they wanted to share, how they wanted to present their digital stories, and how they wanted to make their stories heard. Additional one-on-one guidance was also given to participants via MS Teams, while we guard against over-emphasizing for the participants the process and the product of digital stories [21]. Furthermore, to minimize social desirability in the study, participants were informed that any answer is acceptable in terms of how they practice self-care and were encouraged to be honest. Once participants were ready to submit their digital stories, these were submitted via email, MS Teams, or WhatsApp with the research team only.

The first author (DG) is a post-doctoral fellow trained in qualitative data collection and speaks both isiZulu (the local language of the participants) and English fluently. DG was responsible for data collection. MNS is a full professor and co-investigator in the study, whose responsibility included checking for the accuracy and quality of data. Both DG and MNS were employed at the university; however, they were not involved in teaching the target participants and thus indirectly linked to the participants.

Through thematic analysis of twenty-six digital stories, we elucidated the experiences, understandings, and contexts of the participants and identified potential barriers and facilitators for practicing self-care by first-year students at a South African university. Additionally, researchers held regular debriefing sessions to discuss emerging reflections on ethical and methodological issues arising in the study. We used reflexive journals to keep a record of these challenges including solutions that we implemented to overcome the challenges.

## Ethical considerations

This work was conducted at a university in South Africa. The university's Institutional Research Ethics Committee (Reference Number 009/22) approved this study. Written informed consent was obtained from all participants by completing an online form on MS Teams.

# Results

This section presents the results arising from our reflections on a remote qualitative study that employed the digital storytelling method to explore the self-care practices of first-year students. It is structured into two sub-sections including ethical and methodological challenges and solutions implemented to address the challenges. Fig 2 provides an overview of the challenges encountered and the solutions implemented in our study.

## Ethical challenges encountered with digital storytelling and solutions implemented

Three overarching ethical challenges were identified from our reflections on the research process: (a) online recruitment and informed consent compromised by asynchronous communication, (b) participants' limited research knowledge, and (c) participants' privacy and confidentiality concerns.

**Online recruitment and informed consent compromised by asynchronous communication.** One of the first challenges with the digital storytelling project was a lack of synchronous communication during the online recruitment and informed consenting process. Potential individuals were recruited online by sending out an open invitation to all individuals that met the criteria of the study via email. The email contained a study information letter and an informed consent form. Without synchronous communication, it compromised the ability to engage in a dialogue between the research team and each participant. Therefore, participants emailed questions of clarity or sent mobile phone texts to the research team, and this compromised synchronous communication in answering participants' questions related to the study. Moreover, the nature of the participants were university students. These students were ordinarily committed to online academic studies during the day. As such they opted to check emails in the evening. Those who had questions related to the study emailed or texted the research team in the evening and thus posing another potential for a delay in answering participants' questions.

To compensate for the challenges of asynchronous communication during online recruitment and informed consent processes, the research team adopted three strategies. The first

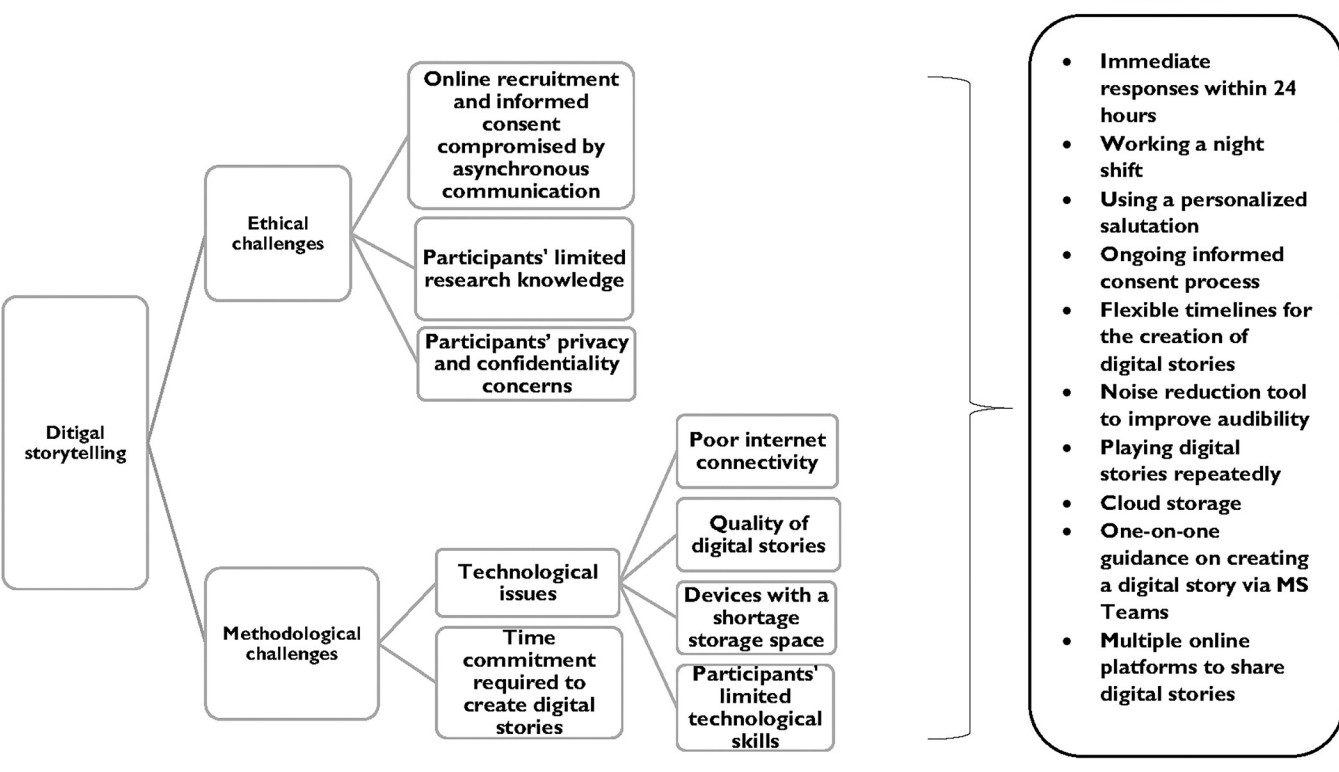

**Fig 2. Overview of the challenges of digital storytelling and the solutions implemented.**

one was ensuring immediate responses within 24 hours to participants who had emailed or texted questions of clarity about the study. The second strategy involved working a night shift in order to be ready to respond to those who opted to contact the research team in the evening. Lastly, using a personalized salutation instead of a generic salutation was also employed when responding to the participants in order to build rapport. All these strategies were effective in compensating for the challenges of asynchronous communication in our digital storytelling project. Although effective, the strategy of working a night shift placed a greater burden on the small research team of two investigators who had to take on extended hours of work. However, personalized salutation and providing potential participants with the immediate information they needed to know about the study to make an informed decision about participating in the research led to an increased likelihood of participation in the context of our digital storytelling project. It also facilitated rapport and strengthened trust between the research team and participants since the participants' questions were addressed timeously. We realized that the remote presence of a consistent, reliable, and responsive research team throughout the project was important for the research participants.

**Participants' limited research knowledge.**   A limited research knowledge of participants posed another challenge in our digital storytelling project. Some participants had a perception that participation in the study was for grades in the university module. For example, one participant asked, "*Will it contribute to my year marks*?" Although it was explained in the study information letter that participation in the study was voluntary and not for grades, it seemed that some participants confused research participation in the digital storytelling project with compulsory academic work due to limited research knowledge. It was clear that the experience

of participating in research was new to the participants. Additionally, the confusion was also exacerbated by not reading the study information letter adequately, as reported by some participants.

As a solution to this, combining the mentioned strategies, namely immediate responses and working a night shift in order to be accessible to participants, with an ongoing process of informed consent was implemented. In other words, participants were immediately contacted to explain the meaning of consent in research and for them to confirm whether they were willing to continue or withdraw participation. In response to the query about grades, it was explained that the digital storytelling project was for research purposes and participation carried no marks or grades. It was clear that the online environment created barriers to assessing participants' full understanding and ability to provide informed consent. However, the ongoing process of informed consent increased participants' understanding of research in general and the informed consent process. The ongoing process of informed consent was effective in ensuring that participants were provided with sufficient information about the study.

**Participants' privacy and confidentiality concerns.** Two participants avoided providing camera exposure to themselves while narrating their self-care practices. One of the participants stated she was shy and uncomfortable showing her face on the camera while narrating her digital story. This participant reported, "*I'm not very comfortable in front of the camera and don't like taking videos of myself.*" Privacy and confidentiality appeared to be an issue for these participants, as they mentioned that they preferred to switch off their cameras while creating the digital stories. Instead, they opted for a voiceover and images to tell their story of how they cared for themselves as first-year students in the era of the COVID-19 pandemic.

In accordance with participants' respect for privacy and ensuring confidentiality in research, participants were permitted to switch off their cameras and assured that none other than the research team would view the digital stories for confidentiality. While we had originally intended for participants to turn on their cameras in order to observe body language, we eventually opted for participants to make individual choices with regard to showing themselves on the camera while narrating their digital stories. This was a simple but effective way to create a safe space for research participants who were not comfortable showing their faces.

## Methodological challenges encountered with digital storytelling and solutions implemented

The digital storytelling method raised a number of technological challenges related to our digital storytelling study of self-care including poor internet connectivity, the quality of self-care digital stories produced by participants, participants' personal devices with a shortage of storage space, and the participants' limited technological skills.

**Poor internet connectivity.** Poor internet connectivity was acknowledged as a methodological challenge of using the internet in research, particularly at the stage whereby participants were ready to submit their digital stories on self-care. Weak signals and unpredictable load-shedding in the country created difficulties for the participants to share files of their digital stories via email or WhatsApp as this needed access to the internet connection. Some participants residing out of university residences and in rural areas reported that slow internet connections required them to move to locations where the internet connection was reliable in order to access emails or WhatsApp. Sometimes they had to stand on top of hills or rocks to access the internet connection for participation in the study. For instance, one participant sent a message to the research team and reported, "*Good morning, I had problems logging in to my Outlook email and MS Teams because of network problems where I'm staying hence replying so late.*" A common expression amongst participants was that they relied only on their mobile phones for

internet access and had no home internet access. Thus, the poor internet connectivity caused disruptions in research participation for some participants.

To address the challenge of poor internet connectivity, participants were granted an extension for the deadline to submit their digital stories on self-care. Given that, upon consenting, participants were requested to create their digital stories and submit them on a given date. The strategy of extending the deadline for submission improved the likelihood for the participants to participate in the study when they knew the deadline was flexible and that the research team accommodated barriers to participation.

**Quality of created digital stories by participants.** The quality of some digital stories submitted by participants was substandard as they contained loud background music and inaudible segments that compromised the voiceover. As a result, the research team struggled to transcribe some digital stories depicting self-care practices that participants adopted during their first year of university.

In order to enhance the transcription and processing of digital stories, we used a noise reduction tool to filter any background sound that compromised the participants' voiceover. In the case of not being able to reduce the background sound, playing digital stories repeatedly to hear the voiceover for data transcription was adopted in this study.

**Devices with a shortage of storage space.** Given that participants were requested to use their personal devices to create digital stories on self-care, some participants reported a shortage of virtual storage space to save their digital stories. Further, we found that restrictions on the size of attachments made it difficult to email files as reported by some participants. As indicated by one participant, "*I tried sending my digital story but it didn't go through.*" While the participants were requested to create short videos not exceeding 10 minutes, some digital stories contained several images and thus increasing the size of the file.

To address this, participants were advised of the various cloud storage platforms to free up room on their devices by using services such as OneDrive, Google Drive, and Dropbox to allow them to save files and make it possible to share larger files of the digital stories with the research team. While this strategy was effective, it also required the research team to be familiar with the technology and troubleshooting procedures to assist the participants.

**Participants' limited technological skills.** While tech-savvy individuals were more experienced and thus more comfortable in creating digital stories, some had limited technological skills, particularly digital media production skills to create digital stories. Moreover, some participants had limited knowledge of how to transfer and share files when they were ready to submit their digital stories to the research team.

In dealing with this challenge, participants were contacted via MS Teams for one-on-one guidance on how to create a digital story and to help address anticipated barriers related to participants' limited digital media production and computer skills. Many participants reported that they found the one-on-one sessions helpful in assisting them with what they were expected to do in creating a digital story. Moreover, participants were given multiple online platforms to submit digital stories through email, WhatsApp, or MS Teams. Further, we extended the submission deadline for those participants who needed additional time to navigate the digital storytelling method until they became familiar and comfortable users of the technology for participation in the study. We found WhatsApp to be a convenient platform and widely available on participants' smartphones to submit digital stories.

**Time commitment required to create digital stories.** One notable advantage of taking this digital storytelling project remotely was that we respected participants' (first-year students) existing academic lives by encouraging them to fit the creation of digital stories into their normal schedule rather than disrupt it. However, it appeared that creating digital stories placed a greater responsibility and time commitment on the participants, who were asked to

complete an additional task of producing digital stories while they had to deal with the academic workload as first-year students. As one participant stated, "*In the meantime, I'm still busy with the assignment and upcoming test. However, I promise that I will send you my digital story by Saturday.*" The majority of participants did not submit digital stories by the specified deadline, reporting a lack of time due to their academic workload.

To manage time challenges in the project, participants were invited to choose any time at their convenience to work on their digital stories while allowing them the flexibility to submit their stories regardless of the requested deadline. As previously indicated, participants were less committed or occupied with routine academic work in the evening.

## Discussion

The Qualitative e-Research Framework [8] helped us to reflect on the ethical and methodological issues arising in our digital storytelling project on students' self-care in the era of the COVID-19 pandemic. As outlined in the reflection above, digital storytelling facilitated engagement with first-year university students during the COVID-19 pandemic. However, successfully recruiting, obtaining informed consent, and collecting data among those who lack knowledge of research and are not tech-savvy present a challenge for health researchers. While qualitative health research remained relevant and vital during the COVID-19 pandemic [6, 7], digital storytelling can help overcome time barriers in the recruitment and data collection processes and enhance the autonomy of research participants in producing research data. Consistent with others [12, 46, 47], remote data collection can minimize the burdens of time and cost of participating in research as both participants and researchers are not required to travel. In the context of university students as research participants, it was convenient to participate in research outside of academic hours. Our study supports evidence from previous research [12] that digital storytelling offered more time for participants to reflect on digital stories prior to sharing them with the research team. However, the reality of challenges to remote qualitative research using digital storytelling with first-year university students in lower-and-middle-income countries (LMIC) settings must be acknowledged and addressed in order to realize the full benefits of the method. Participation in the digital storytelling project was hindered by individual and contextual factors including participants' technological skills; participants' knowledge and experience in public health research; asynchronous communication in the recruitment and data collection processes; concerns for privacy and confidentiality; poor internet connectivity; and participants' technological skills. Despite some of these challenges, digital storytelling provided a valuable opportunity to rise to the challenge of public health restrictions due to the COVID-19 pandemic while ensuring public health research remained relevant during the pandemic.

Other scholars [5, 48] argue that most of the key ethical issues arising in remote qualitative research are similar to those in face-to-face contexts. This differs from our experiences presented here in that there were fundamental differences between ethical issues encountered between remote and in-person qualitative research. Our experiences corroborate the ideas that ethical concerns for remote qualitative research are different from those for face-to-face qualitative research [49]. While both require ethical procedures such as gaining informed consent and ensuring participants' privacy, remote qualitative research, particularly using digital storytelling, requires attention to different issues of obtaining informed consent and the privacy of the participants. For example, remote qualitative work may limit asynchronous communication between the research team and participants during the online recruitment and informed consent processes, thus compromising the ability of dialogue in real time for the participant to ask questions related to the study. Remote qualitative research may afford participants greater

privacy in which participants may have the option to switch off the camera than in face-to-face qualitative work [50]. Furthermore, we do not argue that the solutions presented in this paper cannot be implemented in face-to-face qualitative research or exclusive to digital storytelling. However, they are contextualized for digital storytelling to enable public health researchers to leverage technology to collect data remotely during a pandemic and beyond.

Our experiences in the digital storytelling project are consistent with Lathen and Laestadius [16] in terms of the challenges posed by the limited digital media production and computer skills or by the digital divide. For example, in their study of virtual photovoice, activating the video function was found to be the most common difficulty experienced by participants [14]. Similarly, familiarity with digital media production and computer skills cannot be assumed in qualitative health research with first-year university students in LMIC settings, which is a key reminder of the digital divide considering that many first-year university students in the South African context transition to university with limited computer skills [51]. As suggested by evidence [14], significant time should be spent assisting research participants with basic online functions in online qualitative research to reduce any possible research participation anxiety related to being not tech-savvy. Our experience supports evidence [12] reporting that online data collection such as digital storytelling creates additional burdens and distress for participants who have limited skills in the use of technology. For example, some participants were not competent in using the video function and sharing files. Therefore, the one-on-one technical guidance provided to the research participants was very helpful in building the participants' self-confidence with the methodology and equipping them with new technology skills such as sharing files during our digital storytelling project. However, Archibald and colleagues [52] raise an important point that technical difficulties experienced by participants can also offer an unintended benefit of building rapport and relationship between the researcher and participant as they collaborate in resolving the technical issue. Accordingly, we argue that digital literacy guidance may be needed for online qualitative health studies involving LMIC populations in order to strengthen relationships with participants and minimize technological barriers to participation in public health research.

Another important finding was that some participants created digital stories containing loud music in the background. While loud music compromised the transcription of digital stories, it cannot be associated with the limited digital media production and computer skills of the participants. The behaviour of playing loud music might be associated with the study population of young students. It is possible that participants were excited about digital storytelling as it involved combining music and storytelling. Another possible explanation for this is that previous studies [53, 54] show that listening to music is a strategy used to cope with stress and support well-being among young people. It may be that these participants were trying to project how they coped with stress as young, first-year university students in the midst of the COVID-19 pandemic. In accordance with previous studies [39, 55, 56], university students developed some stressful conditions due to the COVID-19 pandemic, thus listening to music may have been a coping strategy for the COVID-19 pandemic. Despite coping with stress, in digital stories, participants construct narratives and select images and music or sounds they feel readily indicate their experiences [21]. Moreover, consenting without adequately reading the study information letter may suggest poor reading skills and impulsive behaviour of young people.

As pointed out by Ferlatte and colleagues [14], the quality of digital stories due to background music led to inaudible segments in the digital stories and thus challenging accurate transcriptions. We recommend that in future online qualitative studies using digital storytelling need to guide participants to avoid the interference of background noise or loud music when creating digital stories. Although there were very few digital stories that contained

disruptive background music, we recommend that noise reduction tools can be used to reduce background noise during the transcription process.

Several reports [14, 16] have shown that online qualitative research creates barriers to social connectedness and effective interpersonal communications with research participants. In our study, we also experienced a virtual distance between us and the research participants due to physical non-contact during the online recruitment and data collection processes. Pre-COVID-19, qualitative health research often involved an intimate setting with participants during the recruitment and data collection processes. Face-to-face environments were conducive to establishing rapport between the qualitative researchers and the research participants [16]. In our reflection, the remote setting along with asynchronous communication increased demands on the researchers to maintain connection and engagement with the research participants through immediate responses and working a night shift to engage with participants on email and WhatsApp. We believe many of our strategies narrowed the remote distance in the digital storytelling project and motivated the participants to trust the research team as they expressed gratitude for being responsive via email and WhatsApp. The building of trust is regarded as key to the good functioning of a virtual qualitative study for the participants to be open to sharing their experiences, thoughts, and perspectives [14]. It is important for the researchers to safeguard the trust of research participants as it can be broken easily [14], particularly in the context of the COVID-19 pandemic, which was unprecedented and thus anxiety-provoking.

The use of digital storytelling in our project also challenged research participants to stay engaged in the creation of digital stories on their self-care practices while balancing the use of the technology for research participation and for remote teaching and learning. It appears that time management was an issue that threatened the creation of digital stories as the first-year university students juggled with academic workload. Additionally, the timing for online qualitative studies using digital storytelling for data collection needs to be considered so that the data collection processes may not coincide with teaching and learning activities. It is possible that first-year university students may feel pressurized to participate in the digital storytelling project while juggling with remote teaching and learning assessments, thus causing harm to the research participants. The strategy of extending the deadlines for the submission of digital stories may be effective in avoiding any harm to the research participants due to feeling pressurized or anxious to participate in digital storytelling.

It is important to interpret the results of this study in the context in which the study occurred. Firstly, the nature of the study population involved young people transitioning from high school to university. Evidence has shown that first-year university students struggle with the new campus environment [57], and are likely to even confuse voluntary research participation with mandatory academic work. Secondly, first-year university students are often young people transitioning from adolescence to young adulthood and might be exhibiting impulsive behaviours in research participation. Therefore, research studies involving first-year university students need to ensure informed consent is an ongoing process. However, more research is needed to examine the intersection of impulsivity and the decision to consent in research among younger populations. Thirdly, students are committed to academic work during the day and have flexible time to engage in research after hours. Therefore, public health studies involving university students need to align data collection activities to be suitable for this type of study population.

While there were challenges in conducting the digital storytelling project, our view is that the challenges were not the disadvantages of digital storytelling but the challenges of the context in which the study occurred. By the challenges of the research context, we mean asynchronous communication with participants, poor internet connectivity, personal devices with a

shortage of space, the digital divide, and the restrictions imposed by the COVID-19 pandemic. Applying digital storytelling in a different setting may not yield the challenges that we experienced in our study. Therefore, further investigation is needed in a different setting.

## Conclusions

There are several lessons to be learned from our experience regarding the implementation of digital storytelling from a South African perspective. In this article, we contribute to discussions on digital storytelling as an alternative method of data collection by presenting critical reflections on the ethical and methodological challenges of the method and discussing how these challenges can be addressed. We find that, while digital storytelling raises some unique challenges, such as participants' limited technological skills and poor internet connectivity, these challenges may not be issues related to the method but contextual. The context plays a role in creating successful data collection remotely. In our view, the potential for using digital storytelling in public health research is expanding. With the COVID-19 pandemic and beyond, the time and cost of qualitative research, more researchers may opt to use this method. As such, digital storytelling requires further investigation and reflection. In particular, there remains a dearth of practical and ethical guiding principles in qualitative research on this method from the perspectives of participants. Specific participants' experiences with digital storytelling are needed. In addition, more reflection and reflexivity are needed about how technological advancements impact existing digital storytelling procedures and ethics.

## Acknowledgments

We are grateful to all the participants for their contributions to this study.

## Author Contributions

**Conceptualization:** Dumile Gumede, Maureen Nokuthula Sibiya.

**Data curation:** Dumile Gumede, Maureen Nokuthula Sibiya.

**Formal analysis:** Dumile Gumede, Maureen Nokuthula Sibiya.

**Funding acquisition:** Dumile Gumede.

**Methodology:** Dumile Gumede, Maureen Nokuthula Sibiya.

**Project administration:** Dumile Gumede.

**Supervision:** Maureen Nokuthula Sibiya.

**Writing – original draft:** Dumile Gumede.

**Writing – review & editing:** Maureen Nokuthula Sibiya.

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
