## [Decision Letter · Decision Letter 0]

22 Mar 2023

PGPH-D-23-00030

Ethical and methodological reflections: Digital storytelling of self-care with students during the COVID-19 pandemic at a South African University

Dear Dr. Gumede,

Thank you for submitting your manuscript to PLOS Global Public Health. After careful consideration, we feel that it has merit but does not fully meet PLOS Global Public Health’s publication criteria as it currently stands. Therefore, we invite you to submit a revised version of the manuscript that addresses the points raised during the review process.

We look forward to receiving your revised manuscript.

Kind regards,

Lucinda Manda-Taylor, PhD

Academic Editor

Journal Requirements:

Additional Editor Comments (if provided):

Reviewers' comments:

Reviewer's Responses to Questions

**Comments to the Author**

1. Does this manuscript meet PLOS Global Public Health’s publication criteria? Is the manuscript technically sound, and do the data support the conclusions? The manuscript must describe methodologically and ethically rigorous research with conclusions that are appropriately drawn based on the data presented.

Reviewer #1: Partly

Reviewer #2: Yes

2. Has the statistical analysis been performed appropriately and rigorously?

Reviewer #1: N/A

Reviewer #2: N/A

3. Have the authors made all data underlying the findings in their manuscript fully available (please refer to the Data Availability Statement at the start of the manuscript PDF file)?

Reviewer #1: No

Reviewer #2: Yes

4. Is the manuscript presented in an intelligible fashion and written in standard English?

Reviewer #1: Yes

Reviewer #2: Yes

5. Review Comments to the Author

Reviewer #1: This is an interesting paper on digital story telling. My main comment is that the authors must

include more literature on DST in the background e.g why was DST developed, what were the intended goals of DST, what challenges have been experienced elsewhere etc

DiFulvio GT,.(2016), Fiddian-Green A, (2019), de Jager A, (2017), Nyirenda (2022) and Rieger (2018) are some of the references

Methodology

Please have a statement on positionality- indicate who are the data collectors, what are their roles? Are they students or lecturers? Could their social role have an impact on data collected?

How were participants compensated for their time or why were they not compensated?

How did the authors consent for eg family members or friends featured in DST?

How did they plan to support individuals who demonstrated potential for self harm as part of self care?

How did they determine the sample size? How many invitations were sent? How many refusals & how many people did not respond? Did you get more willing students than the required sample size & how was this dealt with? Did they all submit the DST or were there defaulters?

Discussion

What if the study was not done among students- where you can easily access email address? How else can you reach out to hard to reach, marginalised or minority groups in LMICs- in an emergency situation- among who do not even have emails?

Was there any likelihood that first year students would present self care practices that were perceived to be socially acceptable? What if they did?

Other general comments

Lines 78-80- Can authors comment whether people with disabilities eg blind people or socio economically poor people who not have smart phones or ability to read emails can participate in this type of DST?

Line 98 -99- It is not clear what they mean by authors may present. Which authors are they referring to?

Line 113- STIs including HIV & unplanned pregnancies…Please rephrase because it sounds like an unplanned pregnancy is an example of STI

The authors keep using First time entering students and first year students- Can they be consistent? First time entering students is ambiguous because it doesn’t say first time entering what

Line 228-230 Check the grammar & simplify the sentence

Line 405 & several other places- please use LMIC & not LMMC

Reviewer #2: This is a well written paper, setting out clearly the ethical and methodological challenges faced in this study. The ethical challenges encountered are very clearly explained, as are the potential solutions to the challenges. For the purposes of this paper, the authors explain that they are reflecting on the research process. I therefore understood this to be a reflective paper (that focuses on methodological challenges), and not a statistical analysis of results, hence my response to question 2 above (I don't think a statistical analysis is relevant in this case). The paper is wonderfully clear and detailed and can help other researchers who are carrying out remote data collection in a similar context.

6. PLOS authors have the option to publish the peer review history of their article (what does this mean?). If published, this will include your full peer review and any attached files.

**Do you want your identity to be public for this peer review?** For information about this choice, including consent withdrawal, please see our Privacy Policy.

Reviewer #1: No

Reviewer #2: No

---

## [Editor Report · Decision Letter 1]

15 May 2023

Ethical and methodological reflections: Digital storytelling of self-care with students during the COVID-19 pandemic at a South African University

PGPH-D-23-00030R1

Dear Dr Gumede,

We are pleased to inform you that your manuscript 'Ethical and methodological reflections: Digital storytelling of self-care with students during the COVID-19 pandemic at a South African University' has been provisionally accepted for publication in PLOS Global Public Health.

Best regards,

Lucinda Manda-Taylor, PhD

Academic Editor